# Respiration Rate Extraction of Moving Subject Using Velocity Change in FMCW Radar

Jin-Mo Lee , Heemang Song and Hyun-Chool Shin *

Department of Electronic Engineering, Soongsil University, Seoul 156743, Korea; jinmo3982@naver.com (J.-M.L.); ghzero98@gmail.com (H.S.)
* Correspondence: shinhc@ssu.ac.kr

**Abstract:** We propose a novel approach to determine the respiration rate of a moving subject, in terms of the velocity change, by using a frequency-modulated continuous-wave radar. In conventional methods, the respiration rate is determined by considering the variation in the distance between the targets and radar; however, these methods are vulnerable to the subject's movements. The proposed approach estimates the respiration rate by considering the velocity, instead of the distance. An experiment was conducted to measure respiration in several subjects performing various movements. The experimental results demonstrate that the proposed method is more robust to the subject's movements compared to conventional research methods, and can more accurately estimate the respiration rate.

**Keywords:** FMCW radar; vital monitoring; respiration; Doppler velocity; random body movement

## 1. Introduction

In recent times, with the growth in the aging population, people's desire to improve their health is increasing. Consequently, many researchers [1–3] have been conducted to identify methods to periodically monitor an individual's health in daily life and measure vitals to prevent disease. Because respiratory dysfunction can be attributed to cardiovascular, neurological, and psychiatric problems, it is necessary to monitor the vital respiratory signals to identify acute and chronic diseases [4]. In addition, respiratory signals are valuable clinical signs that are included in the clinical risk score pertaining to the National Early Warning Score.

Conventional respiration monitoring methods use a contact sensor. The respiratory signal is extracted from the pressure signal obtained through a wearable belt [1] worn on the body or a single pressure sensor [2] attached to the body. Alternatively, a fiber optic sensor [3] can be attached to the body to measure respiration through the optical interference caused by chest movement.

However, in the sensor contact method, which is commonly used for measuring respiration, the user may feel uncomfortable when the measurement sensor contacts his/her body. To overcome this limitation, respiration measurement methods [5–9] based on frequency-modulated continuous-wave (FMCW) radars are being actively examined. FMCW radars enable the non-contact extraction of distance and displacement information from the target, based on the frequency difference between the transmitting and receiving radio waves. Moreover, because such radars use millimeter waves, the power consumption is low, and the devices can be incorporated into a wide range of applications, owing to the small size of the sensor [10,11]. In addition, in the context of the COVID-19 pandemic, contactless monitoring systems are advantageous for implementing infection control.

In the existing studies [5–9], respiration was measured by considering the magnitude and phase of the radar transmission/receiving demodulation signal only. Notably, even invisible micromovements of the human body that occur while standing without

any support may lead to a distorted signal as the output, owing to the combined signal resulting from the chest movement and other micromovements, which decreases the detection accuracy. Therefore, research [12–15] to monitor respiration in a moving situation is being performed to alleviate the random body movement (RBM) phenomenon that causes distorted signals, and to expand the application range of non-contact respiration methods. Several researchers [12,13] extracted the respiratory frequency, in terms of the modulated frequency shift, by using the motion direction detection as a DC offset, extracted through the phase of the signal that was distorted as a result of the RBM phenomenon. However, because the approach is aimed at predicting the frequency range that is distorted according to the degree of movement, the exact respiratory frequency cannot be determined. Certain researchers [14] used the range-bin alignment method, by combining the bidirectional distance information extracted by two radars, while other researchers [15] discarded the energy that was more than the average value of the respiration signal in a fixed range-bin. Notably, in the method of mitigating the RBM phenomenon through range-bin alignment, the complexity of the structure increases, owing to the use of two radars, and the computational complexity increases, owing to the continuously varying range-bin tracking.

In this study, we deviated from the viewpoint of signal amplitude and phase change associated with chest micromovement caused by respiration. Specifically, we developed a differentiated method for measuring respiration through microvelocity changes according to chest movement. We considered the velocity of vital signals measured with a single channel using the FMCW radar. The extracted velocity information included the velocity frequency associated with the chest movement during respiration. In contrast to the existing methods that track distance changes according to movements, the proposed approach can extract the respiratory frequency by processing the velocity changes caused by chest movements. In addition, the proposed method does not require a measurement range to be set, unlike the existing respiration measurement methods, in which the movement is corrected by setting the range-bin values to alleviate the RBM phenomenon. We conducted four experiments involving movements, and the results demonstrated that the proposed method can more accurately determine the respiration rate in the presence of movements, compared to the existing methods.

## 2. Methods

### 2.1. FMCW Radar Signal Processing

The FMCW radar transmits a linearly modulated radio frequency signal and receives the frequency signal reflected from the target. The combination of the transmitted and received frequency signals is passed through a low-pass filter to extract the signal $x(t, n, m)$ of a target moving at a speed $v_0$. $f_r$ is the frequency corresponding to distance $r$, $M_0$ is the reflected power from $r$, and $P$ is the time delay of the reflected propagation from $r$ [5–7].

$$x(t,n,m) = M_0 \cdot \cos\left(2\pi f_r n + \frac{4\pi \cdot f_c \cdot T_c \cdot v_0}{c} \cdot m + \varphi\right),$$

$$f_r = \frac{2 \cdot BW \cdot \gamma}{c \cdot T_c \cdot F_s},$$

$$P = \frac{4\pi \cdot f_c \cdot T_c \cdot v_o}{c} \cdot m + \varphi, \tag{1}$$

In Equation (1), $m$ denotes the number of chirps, and $n$ is the sample index of the chirps. When calculating $f_r$, $BW$ is the frequency bandwidth, $c$ is the speed of light, $T_c$ is the chirp duration, $\varphi$ is the constant of time delay, $F_s$ is the sampling frequency, and $fc$ is the center frequency.

$P(t, r)$ and $M(t, r)$ were extracted using $X(t, r_k, m)$ via the first-order discrete Fourier transform (DFT) of $x(t, n, m)$.

$$X(t, r_k, m) = \sum_{n=0}^{T_c} x(t, n, m) \cdot e^{-\frac{j \cdot 2\pi \cdot k \cdot n}{T_c}},$$

$$M(t, r_k) = 2|X(t, r_k)| = \frac{M_0}{4\pi \cdot (r_k)^2},$$

$$P(t, r_k) = \angle X(t, r_k) = \frac{4\pi \cdot f_c}{c} \cdot r_k, \tag{2}$$

In Figure 1, $r_k = \frac{c}{2BW}$, and $k = 0, \ldots, N-1$. $\Delta(t)$ represents chest movement owing to respiration.

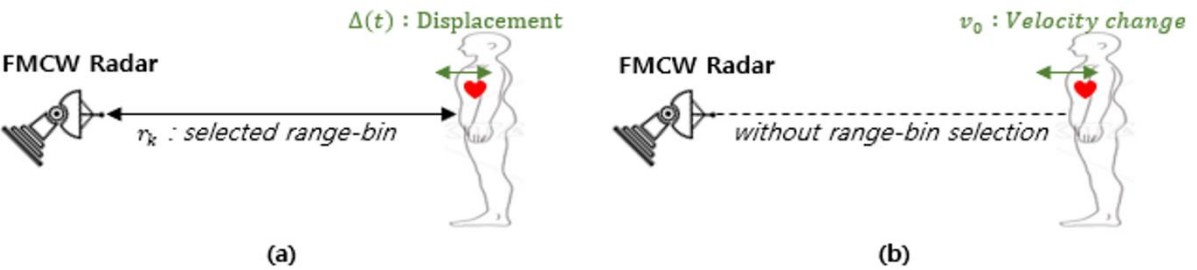

**Figure 1.** Comparison of conventional and proposed approaches. (**a**) Conventional method to measure respiration using the displacement between the radar and subject's chest, which changes owing to the subject's respiration. (**b**) Proposed method to measure respiration using the change in chest velocity, which changes owing to the subject's respiration.

We assume that the object exhibits displacement, owing to the movement of $\Delta(t)$ at $r_k$. $M(t, r_k)$ and $P(t, r_k)$ are expressed as in Equation (3) [5–7,16–18].

$$M(t, r_k) = \frac{M_0}{4\pi \cdot (r_k + \Delta(t))^2},$$

$$P(t, r_k) = \frac{4\pi \cdot f_c}{c} \cdot (r_k + \Delta(t)), \tag{3}$$

Let us explain the basic principle of how the FMCW radar can extract the respiration rate. As shown in Equation (1), the received radar signal in the time domain is given by the following:

$$x(t, n, m) = M_0 \cdot \cos\left(2\pi f_r n + \frac{4\pi \cdot f_c \cdot T_c \cdot v_0}{c} \cdot m + \varphi\right)$$

where $f_r$ is the frequency corresponding to distance $r$, $M_0$. is the reflected power from $r$, and $P$ is the phase component.

$$P = \frac{4\pi \cdot f_c \cdot T_c \cdot v_o}{c} \cdot m + \varphi.$$

This is when the chest (or abdomen) is placed at the distance $r_k$ and exhibits up and down displacement $\Delta(t)$, $M(t, r_k)$, and is expressed as follows [5–7,16–18]:

$$M(t, r_k) = \frac{M_0}{4\pi \cdot (r_k + \Delta(t))^2}.$$

Note that the magnitude component $M(t, r_k)$ in $x(t, n, m)$ is inversely proportional to the chest displacement $\Delta(t)$. As $\Delta(t)$ changes, the magnitude component continuously changes.

Most of the current research detects the respiration rate by exploring $M(t, r_k)$ or $P$. In the figure below, Figure 2a shows the magnitude $M(t, r_k)$ at various distances $r_k$ and Figure 2b shows the magnitude change in $r_k = 0.525$ m according to the chest's up and down movement. As shown in Figure 2, the magnitude $M(t, r_k)$ and $P$ in Figure 2b continuously change, since the reflected power or the time delay of the received radar signal changes. The respiration detection can be obtained by exploring the magnitude or phase change in the target distance.

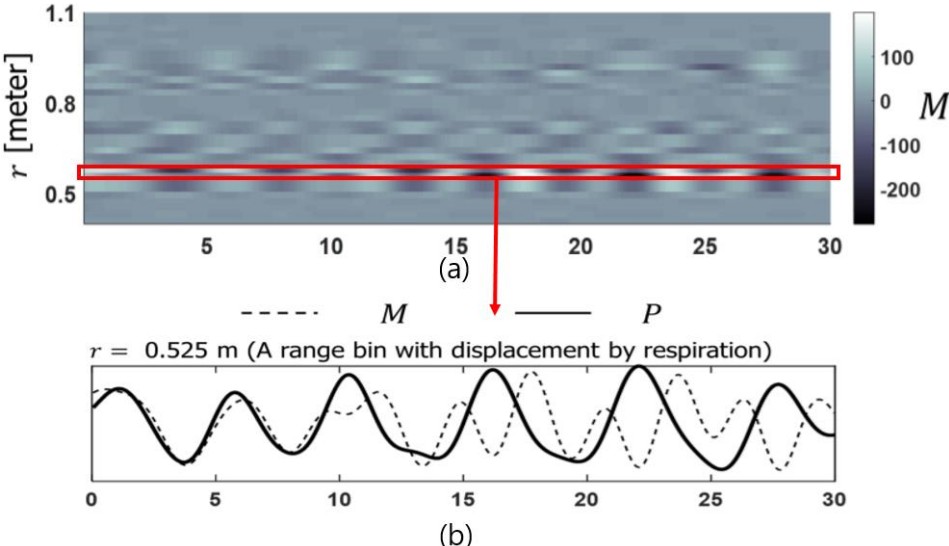

**Figure 2.** Radar signal extracted for respiration measurement. (**a**) The magnitude $M(t, r_k)$ at various distances $r_k$, (**b**) the magnitude change in $r_k = 0.525$ m according to the chest's up and down movement.

An additional experiment was conducted to demonstrate that a respiration measurement is possible with the radar used in this paper. The experiment shown in Figure 3 was conducted using a dummy ("Sakamoto Baby Touch"–Vital Sings Simulator, M179) whose chest displacement, due to respiration, was less than 0.05 m (5 cm). The respiratory rate per minute was set at 20 and the heart rate per minute at 120. The distance between the radar and the dummy was about 0.6 m.

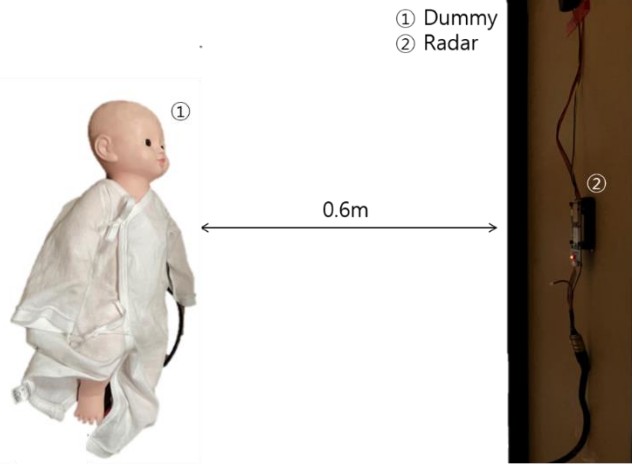

**Figure 3.** Experiment to prove that respiration can be measured using radar.

If $r_k + \Delta(t)$ in Equation (3) includes the displacement pertaining to the movement of the human body and micromovements caused by respiration, distorted $M(t, r_k)$ and $P(t, r_k)$ appear. Therefore, a distorted IF signal is extracted when the subject is in motion.

To extract the velocity to be used in the proposed signal processing method, the second-order DFT is obtained for $X(t, r_k, m)$.

$$Z(t, r_k, v_0) = \sum_{m=0}^{N_{chirp}} X(t, r_k, m) \cdot e^{-\frac{j \cdot 2\pi \cdot k \cdot m}{N_{chirp}}},$$

$$Z(t, r_k, v_0) = \begin{cases} 0, & v \neq v_0 \\ \frac{M_0}{2} \cdot e^{j\varphi}, & v = v_0 \end{cases}, \tag{4}$$

$Z(t, r_k, v_0)$ extracted from Equation (4) only appears in proportion to the reflected power at $r_k$ when the object has a velocity change of $v_0$ at the position $r_k$, as depicted in Figure 1b.

$$D(t, v_0) = \sum_{r_k} |Z(t, r_k, v_0)|, \tag{5}$$

$$D_v(t) = \frac{1}{N_{chirp}} \sum_{v_0} D(t, v_0), \tag{6}$$

In this study, $D(t, v_0)$, obtained through the Doppler map using Equation (5), is substituted into Equation (6) to measure respiration as the average value, in terms of the speed range.

### 2.2. The Study Protocol

The conventional research method, based on $M(t, r_k)$ of data obtained from the FMCW radar, and the proposed method, based on only $D_v(t)$, are compared in four experiments. The FMCW radar used in the experiment has the MOD602 (Bitsensing Inc., Seongnam, Korea) configuration, as shown in Figure 4, and its parameters are listed in Table 1.

$$\Delta R = \frac{c}{2BW} \tag{7}$$

$$\Delta V = \frac{\lambda}{2 \cdot N \cdot T_C} \tag{8}$$

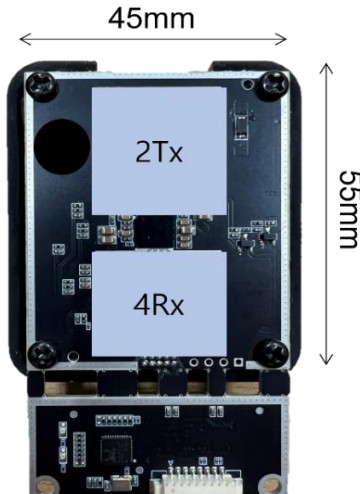

**Figure 4.** FMCW radar.

**Table 1.** Radar parameters and specifications.

| Parameter | Value |
|---|---|
| Center frequency | 60 GHz |
| Detection range | ~6.4 m |
| Bandwidth | 3 GHz |
| Chirp duration | 256 μs |
| Sampling rate | 1 MHz |
| Scan interval | 50 ms |
| Rx antenna spacing | 0.5 λ |
| Range revolution | 0.05 m |
| Velocity revolution | 0.15 m/s |

The number of chirps of the FMCW radar used in this paper is 64. The distance resolution can be calculated by Equation (7), and it is 0.05 m. In addition, the velocity resolution can be obtained using Equation (8), and it is 0.15 m/s. The radar shown in Figure 4 has two transmit antennas and four receive antennas. To evaluate the accuracy of the respiration rate extracted using the radar, the actual respiration rate value is required. To this end, a contact-type respiration sensor (Neulog Respiratory Monitor Belt Logger Sensor; Neulog Inc., Rochester, NY, USA) is used to measure the actual exact respiration rate. The respiratory rate data measured by this sensor are used as the ground truth. To implement the conventional research method, the range-bin corresponding to the subject's chest is selected through the range spectrum, and the respiration rate is measured by considering the chest displacement caused by respiration. In experiment (a), the body is in a stationary state, supported by a wall. Both the conventional and proposed methods can achieve an accurate respiration rate in this stationary situation. In subsequent experiments, as shown in Figure 5b–d, the subject stands still without leaning against a wall, is seated on a chair and moves forward and backward, and is seated on a chair and moves left and right, respectively. We compared the accuracy of the respiration rate extraction using the conventional proposed methods. In all experiments, the FMCW radar is installed approximately 0.6 m in front of the subject's chest, as shown in Figure 6. The respiration signal obtained using the contact sensor in the experiments is defined as the ground truth. In addition, the ground truth signal is compared with the signal extracted using a 0.1–0.4 Hz band-pass filter for the conventional approach, and the signal from which DC is removed is extracted using a 0.1 Hz high-pass filter for the proposed approach. Ten subjects were involved in each experiment and were asked to breathe naturally and normally.

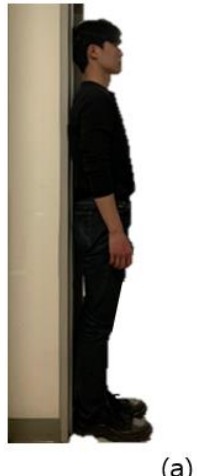 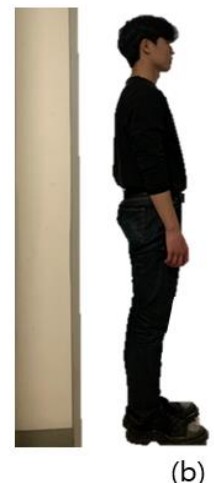 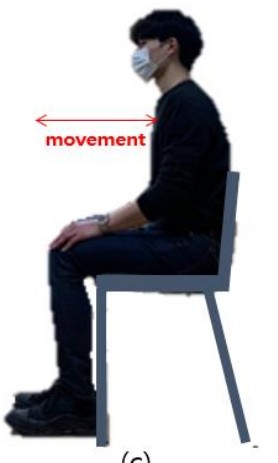 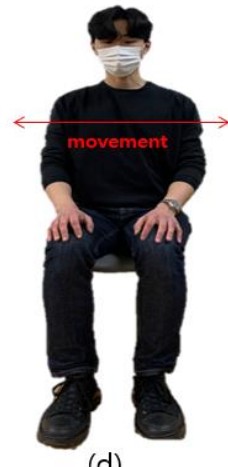

(a) (b) (c) (d)

**Figure 5.** Subject's postures and movements: (**a**) leaning against the wall; (**b**) without leaning against the wall; (**c**) moving forward and backward; (**d**) moving left and right.

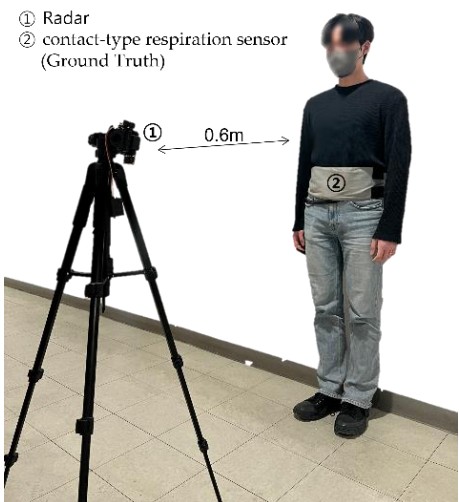

**Figure 6.** Location of subject and radar test equipment.

### 2.3. Comparison of the Conventional Method and the Proposed Method

In the conventional method, even when the subject is standing still while leaning against a wall, a distorted signal is generated, owing to the microscopic movements of the human body, which combine with the displacement caused by the movement of the chest, according to Equation (3). Thus, if the human body is not entirely stationary, the respiration extraction is inaccurate.

In the conventional method, $M(t, r_k)$, inhalation, and exhalation appear in one cycle. However, in the proposed method, $D_v(t)$ exists separately in one cycle each of inhalation and exhalation, according to Equation (5). As depicted in Figure 7, in environment (a), in which the human body is in a stationary state, the outputs of the proposed and conventional methods are accurate respiration signals with a constant cycle. In experimental environments (b), (c), and (d), which include movements other than respiration, the correct respiration signal is extracted from $D_v(t)$. In contrast, when using $M(t, r_k)$, a distorted signal is extracted, owing to the addition of displacement caused by movement other than respiration.

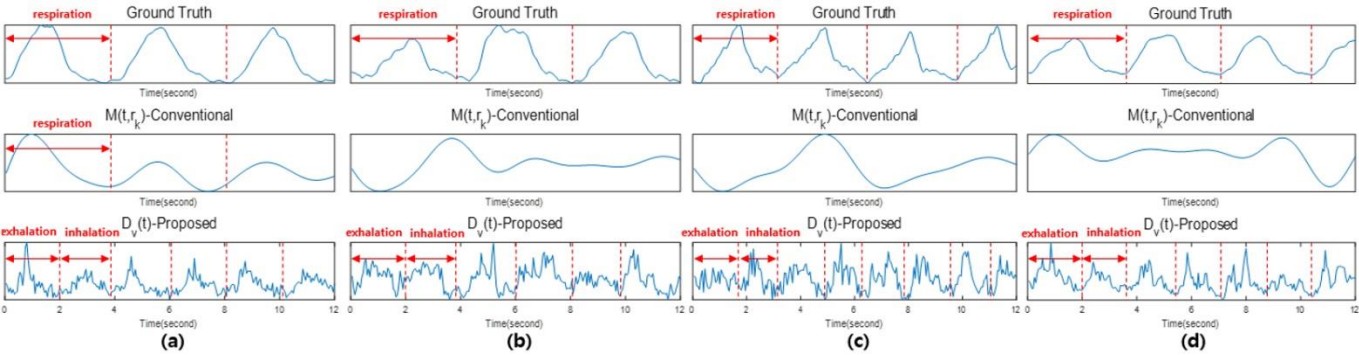

**Figure 7.** Comparison of $D_v(t)$ and $M(t, r_k)$. (**a**) Leaning against the wall; (**b**) without leaning against the wall; (**c**) moving forward and backward; (**d**) moving left and right.

### 2.4. Respiratory Measurement Algorithm

The human respiratory frequency usually lies within the range of 0.1–0.4 Hz. Therefore, if the signal extracted using Equation (6) is divided into windows, and the frequency is extracted through fast Fourier transform (FFT), the respiratory frequency can be extracted within the range of 0.2–0.8 Hz. By comparing the $f_{peak}(t)$ set obtained using FFT and Equation (9), and the previous respiration frequency, $\hat{f}_{tracking}(t-1)$, the peak value closest

to the previous frequency, is determined as $\hat{f}_{tracking}(t)$. The extracted frequency is calculated as the respiratory rate (RR) using Equation (10), and the RR is considered the output.

$$\hat{f}_{tracking}(t) = min\left|\hat{f}_{tracking}(t-1) - f_{peak}(t)\right|, \tag{9}$$

$$RR(t) = \frac{60}{2} \cdot \hat{f}_{tracking}(t), \tag{10}$$

## 3. Results

In this paper, the respiratory rate was extracted in terms of Doppler velocity change. As shown in Equation (4), it is possible to extract the velocity due to the micromovement of the chest during respiration, and the velocity due to movement other than respiration. Based on the data on the previous respiration rate, even in the presence of movement, it is possible to extract a specific speed change frequency by respiration, so the result of suppressing movement was shown.

### 3.1. Respiratory Measurements of Subjects Leaning against the Wall

The respiration rate extraction accuracies of the conventional and proposed methods are compared for ten subjects who leaned against a wall, thereby minimizing the movement of the human body.

As shown in Figure 8, the conventional method yields the same signal cycle as that of the ground truth for a stationary human body, and can accurately determine the RR. The proposed method, in which inhalation and exhalation appear in one cycle, corresponds to two cycles per one cycle of ground truth, and can accurately determine the RR. In this case, both methods use radar to extract the correct values, and the respiration is extracted based on the chest movement associated with respiration, without any other movement of the human body.

$$Accuracy[\%] = \frac{1}{d}\sum_{t=0}^{d}\left(100 - \frac{|RR(t) - G_t(t)|}{G_t(t)} \cdot 100\right) \tag{11}$$

In Equation (11), $d$ is the total experimental time. The RR extraction accuracy is calculated by comparing the ground truth and RR extracted by the two RR extraction methods, using Equation (11). We experimented with the motion conditions shown in Figure 5a. Table 2 shows the respiratory accuracy calculation values for all subjects obtained through the experiment.

**Table 2.** Comparison of respiratory rate accuracy for the case shown in Figure 5a.

| Subject | Conventional | Proposed |
|---|---|---|
| Subject #1 | 99.03% | 99.36% |
| Subject #2 | 90.26% | 93.55% |
| Subject #3 | 94.93% | 98.50% |
| Subject #4 | 76.02% | 92.77% |
| Subject #5 | 85.22% | 95.34% |
| Subject #6 | 76.29% | 94.79% |
| Subject #7 | 77.19% | 91.58% |
| Subject #8 | 78.51% | 84.10% |
| Subject #9 | 80.81% | 92.93% |
| Subject #10 | 82.26% | 95.97% |
| Average | 84.05% | 93.89% |

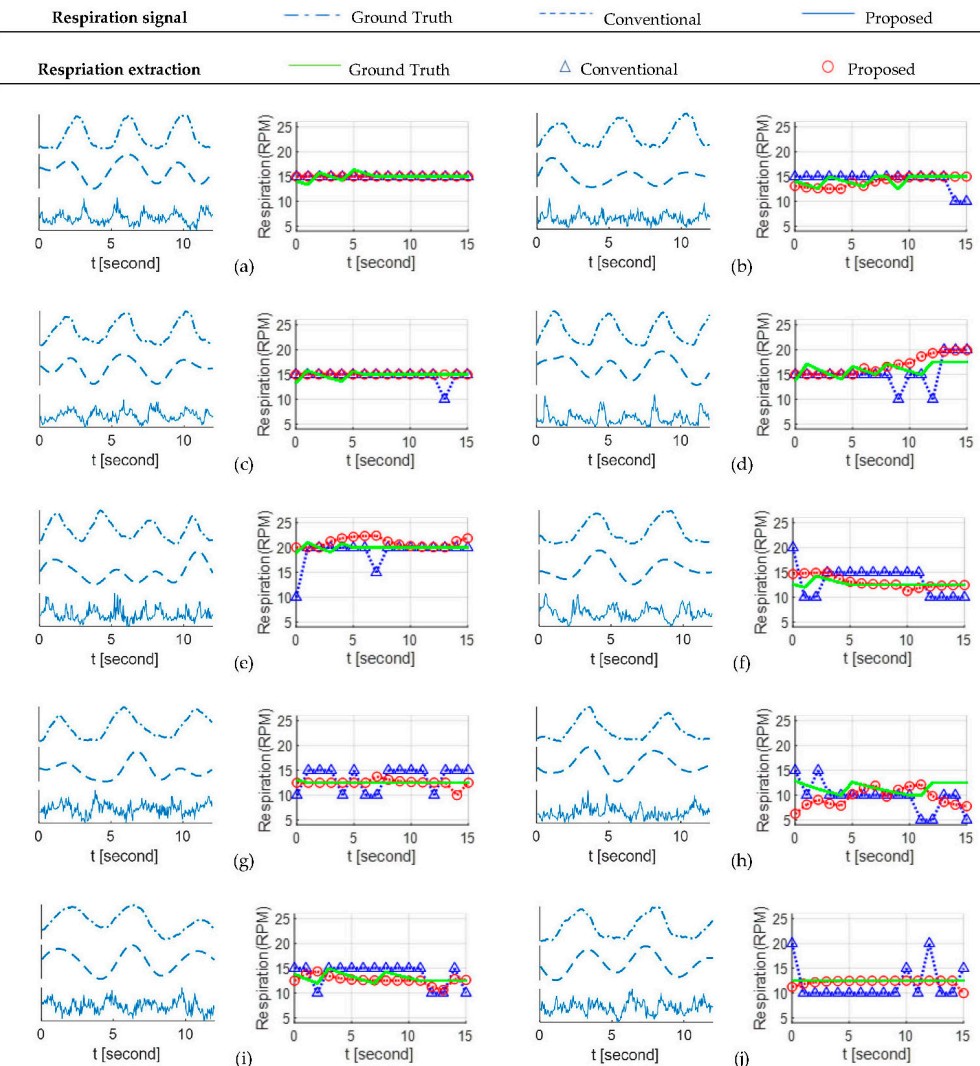

**Figure 8.** Respiration extraction results for the case shown in Figure 5a. (**a**) Subject #1, (**b**) Subject #2, (**c**) Subject #3, (**d**) Subject #4, (**e**) Subject #5, (**f**) Subject #6, (**g**) Subject #7, (**h**) Subject #8, (**i**) Subject #9, and (**j**) Subject #10.

### 3.2. Respiratory Measurements of Subjects without Leaning against the Wall

The respiration rate extraction accuracies of the conventional and proposed methods are compared for ten subjects who stood upright without leaning against the wall.

As shown in Figure 9, the conventional method cannot accurately determine the respiration rate, owing to the added displacement pertaining to movement other than chest movement. However, because the proposed method can determine the specific velocity change in the chest due to respiration, a signal can be generated for each inhalation and exhalation cycle in one cycle, leading to accurate extraction of the respiration rate. We experimented with the motion conditions shown in Figure 5b. Table 3 shows the respiratory accuracy calculation values for all subjects obtained through the experiment.

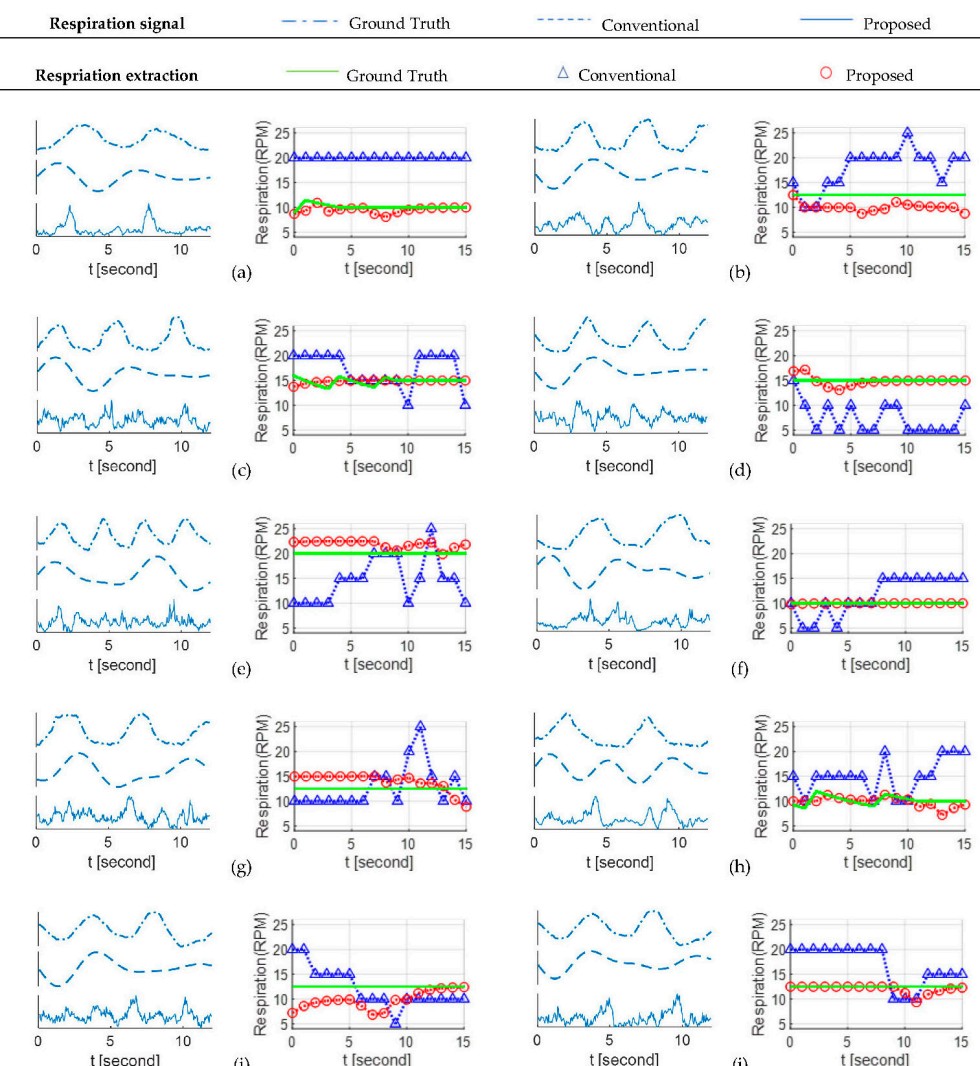

**Figure 9.** Respiration extraction results for the case shown in Figure 5b. (**a**) Subject #1, (**b**) Subject #2, (**c**) Subject #3, (**d**) Subject #4, (**e**) Subject #5, (**f**) Subject #6, (**g**) Subject #7, (**h**) Subject #8, (**i**) Subject #9, and (**j**) Subject #10.

**Table 3.** Comparison of respiratory rate accuracy for the case shown in Figure 5b.

| Subject | Conventional | Proposed |
|---|---|---|
| Subject #1 | 64.56% | 92.97% |
| Subject #2 | 72.80% | 81.62% |
| Subject #3 | 86.36% | 97.08% |
| Subject #4 | 73.57% | 90.81% |
| Subject #5 | 77.68% | 89.14% |
| Subject #6 | 66.56% | 75.05% |
| Subject #7 | 70.32% | 80.92% |
| Subject #8 | 51.57% | 87.26% |
| Subject #9 | 69.88% | 73.97% |
| Subject #10 | 50.40% | 90.38% |
| Average | 68.37% | 85.92% |

### 3.3. Respiratory Measurements of Subjects Moving Forward and Backward

The respiration rate extraction accuracies of the conventional and proposed research methods are compared for ten subjects seated in a chair and moving forward and backward.

As shown in Figure 10, the conventional method alleviates the effect of chest movement displacement caused by respiration, owing to the change in distance from the radar. Nevertheless, the result is inaccurate because the signal beyond the set range-bin is measured. In contrast, the proposed method can extract the specific velocity change in the chest owing to respiration, regardless of the measurement range. In addition, because the velocity change frequency associated with movement other than chest movement exists individually, the respiration rate can be accurately determined. We experimented with the motion conditions shown in Figure 5c. Table 4 shows the respiratory accuracy calculation values for all subjects obtained through the experiment.

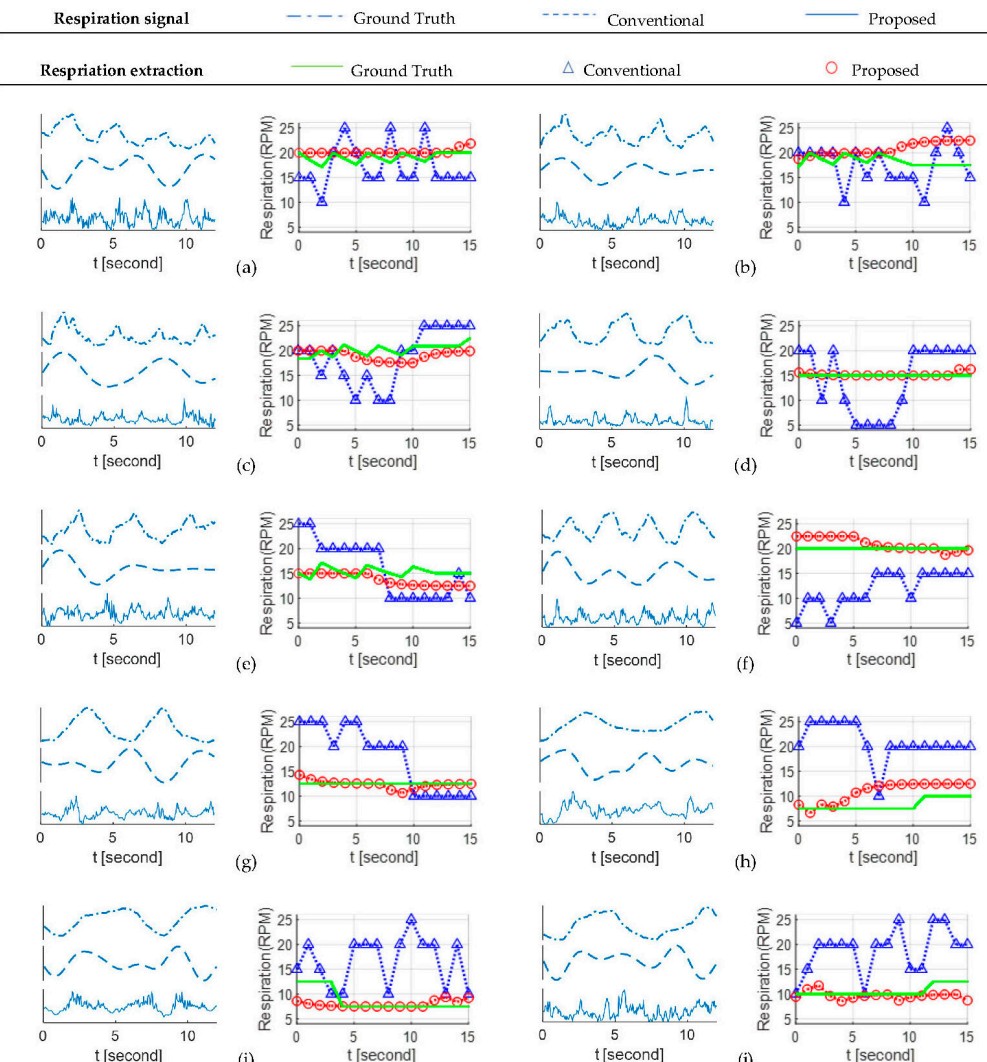

**Figure 10.** Respiration extraction results for the case shown in Figure 5c. (**a**) Subject #1, (**b**) Subject #2, (**c**) Subject #3, (**d**) Subject #4, (**e**) Subject #5, (**f**) Subject #6, (**g**) Subject #7, (**h**) Subject #8, (**i**) Subject #9, and (**j**) Subject #10.

**Table 4.** Comparison of respiratory rate accuracy for the case shown in Figure 5c.

| Subject | Conventional | Proposed |
|---|---|---|
| Subject #1 | 69.62% | 89.57% |
| Subject #2 | 69.71% | 84.06% |
| Subject #3 | 71.13% | 93.98% |
| Subject #4 | 80.95% | 88.14% |
| Subject #5 | 59.57% | 85.97% |
| Subject #6 | 65.59% | 92.66% |
| Subject #7 | 47.46% | 89.53% |
| Subject #8 | 58.85% | 80.77% |
| Subject #9 | 41.72% | 83.52% |
| Subject #10 | 50.11% | 83.19% |
| Average | 61.47% | 87.13% |

### 3.4. Respiratory Measurements of Subjects Moving Left and Right

The respiration rate extraction accuracies of the conventional and proposed methods are compared for ten subjects seated in a chair and moving left and right.

As depicted in Figure 11, the conventional method alleviates the effect of displacement of the chest movement associated with respiration, owing to the change in distance from the radar. Nevertheless, the result is inaccurate because the signal beyond the set range-bin is measured. In contrast, the proposed method can extract the specific velocity change in the chest owing to respiration, regardless of the measurement range. In addition, because the velocity change frequency associated with movement other than chest movement exists individually, the respiration rate can be accurately determined. We experimented with the motion conditions shown in Figure 5d. Table 5 shows the respiratory accuracy calculation values for all subjects obtained through the experiment.

**Table 5.** Comparison of respiratory rate accuracy for the case shown in Figure 5d.

| Subject | Conventional | Proposed |
|---|---|---|
| Subject #1 | 84.66% | 94.63% |
| Subject #2 | 67.09% | 93.04% |
| Subject #3 | 61.09% | 86.12% |
| Subject #4 | 49.62% | 88.45% |
| Subject #5 | 74.14% | 87.49% |
| Subject #6 | 56.91% | 78.14% |
| Subject #7 | 64.76% | 80.63% |
| Subject #8 | 52.39% | 87.92% |
| Subject #9 | 72.21% | 87.71% |
| Subject #10 | 51.04% | 74.25% |
| Average | 63.39% | 85.84% |

Since the proposed method outputs the respiration rate every second and the processing time is about 0.2–0.4 s, the method is sufficiently capable of real-time implementation.

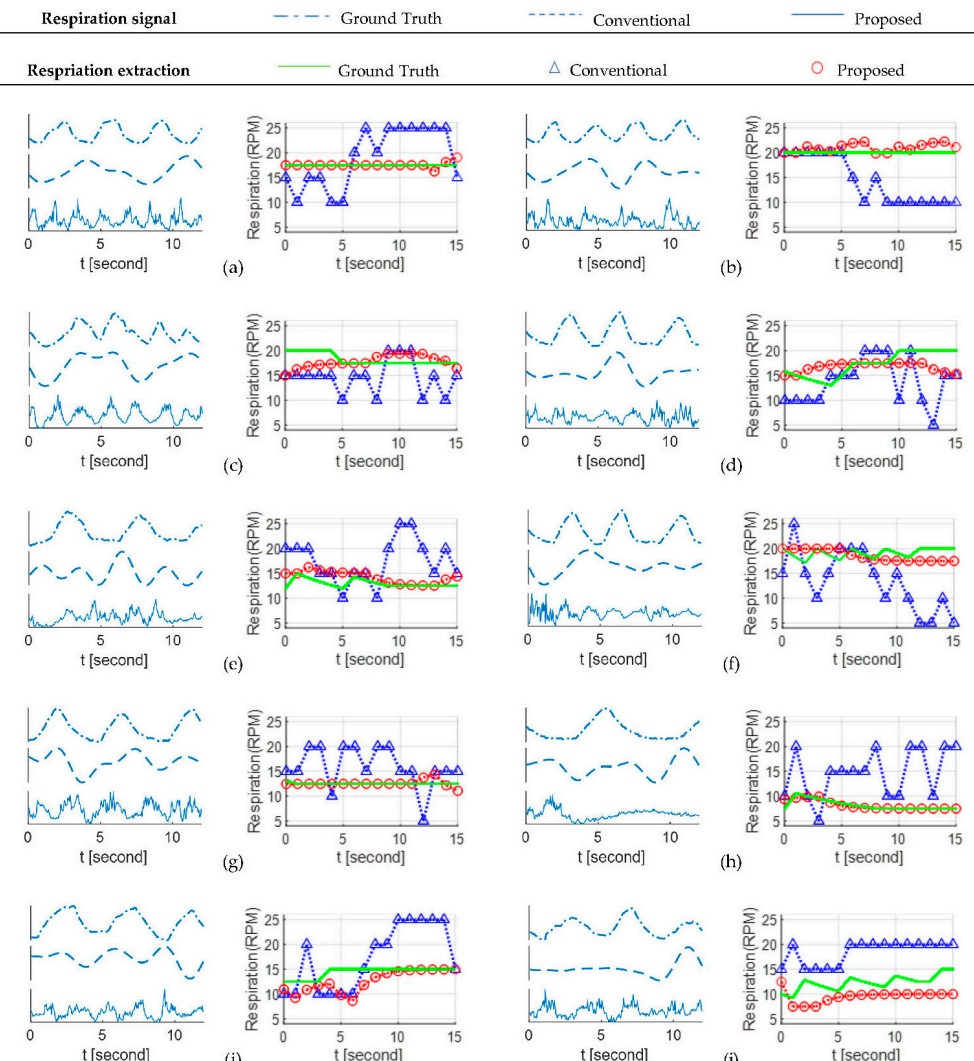

**Figure 11.** Respiration extraction results for the case shown in Figure 5d. (**a**) Subject #1, (**b**) Subject #2, (**c**) Subject #3, (**d**) Subject #4, (**e**) Subject #5, (**f**) Subject #6, (**g**) Subject #7, (**h**) Subject #8, (**i**) Subject #9, and (**j**) Subject #10.

### 3.5. Correlation Analysis of Conventional and Proposed Method

To statistically analyze the performance of the conventional and proposed methods, a correlation analysis was performed. In Equation (12), $\rho$ measures the linear correlation between two sets of values:

$$\rho = \frac{\sum_i \left(RR_i - \overline{RR}\right) \cdot \left(G_{ti} - \overline{G_t}\right)}{\sigma_{RR} \cdot \sigma_{G_t}} \tag{12}$$

where $RR$ is the extracted respiration rate by the radar sensor and $G_t$, is the ground truth.

Table 6 shows the correlation analysis of the four experiments presented in Figure 5. The correlation coefficient of the proposed method is higher than that of the conventional method. This means that the extracted respiration by the proposed method is closer to the actual respiration rate.

**Table 6.** Correlation analysis of respiratory rate extracted from 4 experiments.

| Method | $\rho$ of Figure 5a | $\rho$ of Figure 5b | $\rho$ of Figure 5c | $\rho$ of Figure 5d |
|---|---|---|---|---|
| Conventional | 0.5276 | 0.1907 | $-0.1561$ | 0.1514 |
| Proposed | 0.8805 | 0.8470 | 0.8375 | 0.6362 |

## 4. Conclusions

We proposed a novel method of extracting the respiratory rate by investigating the change in chest velocity caused by respiration. Conventional methods have extracted the respiration rate by measuring the movement displacement of the chest due to respiration, and have resulted in inaccurate outputs when subjects perform movements other than respiration. This performance degradation is mainly because the target range-bin is changed due to chest movement. In the proposed method, a signal processing technique for extracting velocity change in the chest was used to overcome the limitation. The experimental results showed that the correlation coefficient of the proposed method is significantly higher than that of conventional studies.

However, the proposed study may still result in an inaccurate respiratory rate when the velocity change due to body movement is much greater than the chest velocity change due to respiration. In addition, the cases in which the radar is placed in various positions, such as the side or the rear of the subject's chest, rather than the front, need to be investigated.

Since the proposed method is simple enough to be implemented in real time, it can be used in various fields, such as extracting the driver's respiration rate in a vehicle or extracting the respiration rate of a moving person during sleep.

**Author Contributions:** Conceptualization, J.-M.L., H.S. and H.-C.S.; Data curation, J.-M.L.; Formal analysis, J.-M.L.; Funding acquisition, H.-C.S.; Investigation, H.-C.S.; Methodology, J.-M.L. and H.S.; Project administration, H.S.; Resources, H.-C.S.; Software, J.-M.L.; Supervision, H.-C.S.; Validation, H.S.; Visualization, J.-M.L.; Writing—original draft, J.-M.L.; Writing—review & editing, H.S. and H.-C.S. All authors have read and agreed to the published version of the manuscript.

**Funding:** This research was supported by a grant of the Korea Health Technology R&D Project through the Korea Health Industry Development Institute (KHIDI), funded by the Ministry of Health & Welfare, Republic of Korea (grant number: HI21C0852).

**Institutional Review Board Statement:** Not applicable.

**Informed Consent Statement:** Informed consent was obtained from all subjects involved in the study.

**Data Availability Statement:** Not applicable.

**Conflicts of Interest:** The authors declare no conflict of interest.

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
