# Peer review of "Respiration Rate Extraction of Moving Subject Using Velocity Change in FMCW Radar"

_applsci, doi:10.3390/app12094114_

Round 1

Reviewer 1 Report

The authors provided a new method to detect the respiration rate excluding the body movement’s disturbance. And the results showed the effects. However,some analysis should be presented more clearly 1、 What is the difference between this approach and multi-frequency FSK? How many Nchirp is needed to validate the 2、 The authors should explain how the body movement is suppressed explicitly in the results. The figures6-9 could be simplified. The results could be presented more efficiently. The equation 1 missed some parameters’definitions like γ(gamma),fc?

Author Response

Dear reviewer,

Thank you for valuable comments, and an opportunity to revise the paper.

We are uploading our point-by-point response to the comments (below) (response to reviewers), and an updated manuscript.

Best regards,

Hyun-Chool Shin

Reviewer 2 Report

The authors proposed a method of extracting the respiration signals from the radar echo without the need of finding the proper range bin. This is very interesting.  But there are still some parts making me in a confusion.

  1. Section 2.1 is the core of your method, but I cannot understand eq. 1 well as there is no variable 't' in the right hand. I guess that you want to use t=r/c to replace a simple variable 't'. Also,  in the second row of eq. 1, it is gamma or r?
  2. Line 87, please introduce M(t,r) and P(t,r) for an easy understanding.
  3. How do you sync your measurement data with the reference?
  4. The conventional method is not quite clear, and it is better to add the comparison with the state-of-the-art work in some publications.

Author Response

(The authors gave the same response as above.)

Reviewer 3 Report

The authors presented a very interesting work. The work has good scientific soundness and technically well-presented. This reviewer recommend to accept this manuscript after addressing the following minor comment:

1. Please, clearly refer and explain the "conventional" method used/mentioned in the manuscript, such as exactly what conventional method was used and also explain the mechanism/algorithm of the conventional method used in the study.

Author Response

(The authors gave the same response as above.)

Reviewer 4 Report

The paper proposes a novel approach to measure respiratory rate (RR) based on Frequency Modulation Continuous Wave (FMCW) radar. The novel method is based on measuring the chest velocity instead of the displacement to reduce distortion. The RR obtained with the  novel and the conventional FMCW radar methods have been tested against the contact-type respiration sensor and results show an improvement of the accuracy of measurements.

The paper is of interest and innovative. The aim of the work is clearly exposed. Some improvements are required to the Methods and Results sections.

Report in the abstract the tests you performed to test the new method

The paragraph 2.3 should be moved to the Results section

The paragraph 3.5 should be moved in the Methods section

Please move the Figure 2 and Figure 3 to the Study protocol section

Author Response

(The authors gave the same response as above.)

Round 2

Reviewer 1 Report

The present form is acceptable.

The 'r' in equation1 cannot be recognized. Is it because of font?

Reviewer 4 Report

the authors addressed the comments of the previous revision and the manuscript is suitable for publication in the present form